



# On the annual variability of Antarctic aerosol size distributions at Halley research station

Thomas Lachlan-Cope[1], David Beddows[2], Neil Brough[1], Anna E.
Jones[1], Roy M. Harrison[2,+], Angelo Lupi[3], Young Jun Yoon[4], Aki
Virkkula[5,6] and Manuel Dall´Osto[7*]
*[1]British Antarctic Survey, NERC, High Cross, Madingley Rd, Cambridge, CB3*
*0ET, United Kingdom*
*[2]National Center for Atmospheric Sciences, University of Birmingham,*
*Edgbaston, Birmingham, B15 2TT, United Kingdom*
*[3]Institute of Atmospheric Sciences and Climate (ISAC),National Research*
*Council (CNR), via P. Gobetti 101, 40129, Bologna Italy*
*[4]Korea Polar Research Institute, 26, SongdoMirae-ro, Yeonsu-Gu, Incheon,*
*KOREA 406-840*
*[5]Institute for Atmospheric and Earth System Research, University of Helsinki*
*Helsinki, FI-00014, Finland*
*[6]Finnish Meteorological Institute, FI-00101 Helsinki, Finland*
*[7]Institute of Marine Sciences,* Passeig Marítim de la Barceloneta, 37-49. E-
08003, Barcelona, Spain; corresponding author, email: dallosto@icm.csic.es
[+]Also at: Department of Environmental Sciences/Centre of Excellence in
Environmental Studies, King Abdulaziz University, PO Box 80203, Jeddah,
21589, Saudi Arabia.



**Abstract**
The Southern Ocean and Antarctic region currently best represent one of the
few places left on our planet with conditions similar to the preindustrial age.
Currently, climate models have low ability to simulate conditions forming the
aerosol baseline; a major uncertainty comes from the lack of understanding of
aerosol size distributions and their dynamics. Contrasting studies stress that
primary sea-salt aerosol can contribute significantly to the aerosol population,
challenging the concept of climate biogenic regulation by new particle
formation (NPF) from dimethyl sulphide marine emissions.
We present a statistical cluster analysis of the physical characteristics of
particle size distributions (PSD) collected at Halley (Antarctica) for the year
2015 (89% data coverage). By applying the Hartigan-Wong k-Means method
we find 8 clusters describing the entire aerosol population. Three clusters
show *pristine* average low particle number concentrations (< 121-179 $cm^{-3}$)
with three main modes (30 nm, 75-95 nm, 135-160 nm) and represent 57% of
the annual PSD (up to 89-100% during winter, 34-65% during summer based
upon monthly averages). Nucleation and Aitken mode PSD clusters dominate
summer months (Sep-Jan, 59-90%), whereas a clear bimodal distribution (43
and 134 nm, respectively,  min Hoppel mode 75 nm) is seen only during the
Dec-Apr period (6-21%). Major findings of the current work include: (1) NPF
and growth events originate from both the sea ice marginal zone and the
Antarctic plateau, strongly suggesting multiple vertical origins, including
marine boundary layer and free troposphere; (2) very low particle number
concentrations are detected for a substantial part of the year (57%), including
summer (34-65%), suggesting that the strong annual aerosol concentration
cycle is driven by a short temporal interval of strong NPF events; (3) a unique
pristine aerosol cluster is seen with a bimodal size distribution (75 nm and 160
nm, respectively), strongly correlating with wind speed and possibly
associated with blowing snow and sea spray sea salt, dominating the winter
aerosol population (34-54%). A brief comparison with two other stations
(Dome C Concordia and King Sejong Station) during the year 2015 (240 days
overlap) shows that the dynamics of aerosol number concentrations and



distributions are more complex than the simple sulphate-sea spray binary combination, and it is likely that an array of additional chemical components and processes drive the aerosol population. A conceptual illustration is proposed indicating the various atmospheric processes related to the Antarctic aerosols, with particular emphasis on the origin of new particle formation and growth.

## 1 Introduction

Atmospheric marine aerosol particles contribute substantially to the global aerosol budget; they can impact the planetary albedo and climate (Reddington et al., 2017). However, aerosols remain the least understood and constrained aspect of the climate system (Boucher et al., 2013). Aerosol concentration, size distribution, chemical composition and dynamic behavior in the atmosphere play a crucial role in governing radiation transfer. However, aerosol sources and processes, including critical climate feedback mechanisms, are still not fully characterized. This is especially true in pristine environments, where the largest uncertainties are found, mainly due to lack of understanding of pristine natural sources (Carslaw et al., 2013). Indeed, the Southern Ocean and the Antarctic region still raises many unanswered atmospheric science questions.  This region has complex interconnected environmental systems - such as ocean circulation, sea ice, land and snow cover – which are very sensitive to climate change (Chen et al., 2009).

Early research upon Antarctic aerosols was carried out over various part of the continent  and reviewed by Shaw et al. (1988). It was concluded that a peculiar feature of the Antarctic aerosol system is a very pronounced annual cycle of the total particle number concentration, with concentrations 20-100 times higher during austral summer than during winter.

This seasonal cycle - like a seasonal "pulse" over the summer months (December, January and February) - seems to be more prominent in the upper Antarctic plateau than the coastal Antarctic zones, but particle number concentrations are much higher in coastal Antarctica. One possible origin for these nuclei could be the Antarctic free troposphere, as suggested by Ito et al.



(1993), although this free troposphere to marine boundary layer transport was
considered by no means a definite explanation (Koponen et al., 2002; 2003).
Overall, the aerosol summer maximum concentrations can be largely
explained by new particle formation (NPF) events, as recently reviewed by
Kerminen et al., (2018).
The vertical origin of these NPF events is still matter of debate. Some
indications suggesting NPF takes place preferentially in the Antarctic Free
Troposphere (FT): aerosols originate in the upper troposphere, then the
circulation induced by the Antarctic drainage flow (James, 1989) transports
aerosols down to the boundary layer in the Antarctic plateau, with subsequent
transport further to the coast by katabatic winds (Ito et al., 1993; Koponen et
al., 2002; Fiebig et al., 2014; Hara et al., 2011; Järvinen  et al., 2013;
Humphries et al., 2016). A recent study found that the Southern Ocean was
the dominant source region for particles observed at Princess Elisabeth (PE)
station, leading to an enhancement in particle number (N), while the Antarctic
continent itself was not acting as a particle source (Herenz et al., 2019).
Further studies also point to boundary layer oceanic sources of NPF events
(Weller et al., 2011; Weller et al., 2015; Weller et al., 2018). Recently, a long
term analysis of the seasonal variability in the physical characteristics of
aerosol particles sampled from the King Sejong Station (located on King
George Island at the top of the Antarctic Peninsula) was reported (Kim et al.,
2017). The CCN concentration during the NPF period increased by
approximately 11 % compared with the background concentration (Kim et al.,
2019). Interestingly, new particle formation events were more frequent in the
air masses that originated from the Bellingshausen Sea than in those that
originated from the Weddell Sea, and it was argued that the taxonomic
composition of phytoplankton could affect the formation of boundary layer new
particles in the Antarctic Ocean (Jang et al., 2019). Dall´Osto et al. (2017)
reported higher N in sea ice-influenced air masses.
Overall, studies to date suggest that regional NPF events in Antarctica are not
as frequent as those in the Arctic or other natural environments, although the
growth rates are similar (Kerminen et al., 2018). In terms of aerosol size, most
of the ultrafine (<100 nm) particle concentrations have been linked to NPF



events, whereas sea salt particles dominate the coarse mode and
accumulation mode (>100 nm). A recent study by Yang et al. (2019), however,
proposes a source for ultrafine sea salt aerosol particle from blowing snow,
dependent on snow salinity. This mechanism could account for the small
particles seen during Antarctic winter at coastal stations.
It is interesting to note that the recent, spatially-extensive study of the
concentration of sea-salt aerosol throughout most of the depth of the
troposphere and over a wide range of latitudes (Murphy et al., 2019) reported
a source of sea-salt aerosol over pack ice that is distinct from that over open
water, likely produced by blowing snow over sea ice (Huang et al., 2018;
Giordano et al., 2018; Frey et al., 2019). In recent years, a number of long
term aerosol size distribution datasets have been discussed (Järvinen et al.,
2013; Kim et al., 2019) but these types of datasets are still scarce. The ability
to measure aerosol size distributions at high time resolution allows open
questions to be investigated. The purpose of the present work is to examine
for the first time a one year long (2015) dataset collected at Halley Station.
Previous work at the Halley research station reported size-segregated aerosol
samples collected with a cascade impactor at 2 week intervals for a year. Sea
salt was found to be a major component of aerosol throughout the year (60%
of mass) deriving from the sea ice surface rather than open water.
Methanesulphonic Acid (MSA) and non-sea-salt sulphate both peaked in the
summer and were found predominantly in the submicron size range (Rankin
and Wolff, 2003). Observations of new particle formation during a two month
cruise in the Weddell Sea revealed an iodine source (Atkinson et al., 2012).
While no short-term correlation (timescale < 2 days) was found between
particles and iodine compounds in a later study (Roscoe et al., 2015), the
authors highlighted correlations on seasonal timescales. It is also worth
mentioning that a previous Weddell Sea study also found increased new
particle formation in the sea ice zone (Davison et al., 1996), but no clear
correlation between dimethyl sulphide and new particle bursts was found.





In this paper, we use k-means cluster analysis (Beddows et al., 2009) to
elucidate the properties of the aerosol size distributions collected across the
year 2015 at Halley.   A clear advantage of this clustering method over
average size distributions (e.g. monthly, seasonally, etc.) is that specific
aerosol categories of PSD can be compared across different time periods.
While a number of intensive polar field studies have focused on average
monthly datasets, cluster analyses of year long polar and marine particle size
distributions measurements are scarce. Recently, cluster analysis was applied
to Arctic aerosol size distributions taken at Zeppelin Mountain Svalbard;
Dall'Osto et al., 2017a) during an 11-year record (2000–2010) and at Villum
Research Station (Greenland; Dall'Osto et al., 2018b) during a 5-year period
(2012–2016).  Both studies showed a striking negative correlation between
sea ice extent and nucleation events, and concluded that NPF are events
linked to biogenic precursors released by open water and melting sea ice
regions, especially during the summer season. Recently, data from three high
Arctic sites (Zeppelin research station, Gruvebadet Observatory, Villum
Research Station at Station Nord) over a 3-year period (2013–2015) were
analysed via clustering analysis, reporting different categories including
pristine low concentrations (12 %–14 % occurrence), new particle formation
(16 %–32 %), Aitken (21 %–35 %) and accumulation (20 %–50 %) particles
categories (Dall´Osto et al., 2019). To our knowledge, this is the first year-long
Antarctic dataset where cluster analysis has been applied. The objective of
this work is to analyze different types of aerosol size distributions collected
over a whole year of measurements, to elucidate source regions (including
open ocean, land, snow on land, consolidated and marginal sea ice zones),
discuss possible primary and secondary aerosol components, and propose
mechanisms where NPF and growth may take place in the study region.





## 2. Methods

### 2.1 Location

The measurements reported here were made at the British Antarctic Survey's Halley VI station (75° 36'S, 26° 11'W), located in coastal Antarctica, on the floating Brunt Ice Shelf ~20 km from the coast of the Weddell Sea. A variety of measurements were made from the Clean Air Sector Laboratory (CASLab), which is located about 1 km south-east of the station (Jones et al., 2008).

### 2.2 SMPS and CPC

The aerosol size distribution was measured using a TSI Inc. Scanning Mobility Particle Sizer (SMPS), comprising an Electrostatic Classifier (model 3082), a Condensation Particle Counter (CPC) model 3775, and a long Differential Mobility Analyser (DMA, model 3081). The SMPS returned information on numbers of particles in discrete size bins in the size range 6 nm to 209 nm, at 1-min temporal resolution. A condensation particle counter (CPC, TSI Inc. model 3010) is routinely run at Halley. It provides a measure of total number of particles with diameter between 10 nm and ~3 microns. Both instruments sampled from the CASLab's central, isokinetic, aerosol stack (200 mm i.d. stainless steel) (see Jones et al. (2008) for details).

### 2.2.1. SMPS K means clustering data analysis

Prior to clustering, the SMPS distributions are normalized so that the Euclidean length of each (treated as a vector) is 1. This ensures that we are clustering the shape of the distributions irrespective of the magnitude of the number count within each. The normalized data given then are clustered using the k-means (method R Core Team (2019). This partitions the SMPS distributions (treated as vectors by k-means) into k groups such that the sum of squares of the distances from these points to the assigned cluster centres





is minimized.  At the minimum, the cluster centres form the average SMPS
distributions of the individual SMPS distributions assigned to each cluster.
To decide on the number of factors to choose, the Dunn Index and Silhouette
Width were calculated for each factor number.  The Dunn Index is the ratio of
the smallest distance between observations not in the same cluster to the
largest intra-cluster distance. The Dunn Index has a value between zero and
infinity, and should be maximized. Similarly, the Silhouette Width analysis is a
measure of how similar the observations are with the cluster they are
assigned to relative to other clusters. Its value ranges from -1 to 1 for each
observation in your data.  A value approaching 1 indicates that the elements
within each cluster are identical to each other; a values close to 0 suggest that
there is no clear division between clusters; and a value to -1 suggest that the
observations have been assigned to the wrong cluster.  As we increase the
cluster number from 2 up to 30 the Silhouette Width falls from a maximum
value of 0.49 to 0.28 and the Dunn Index increases from a minimum of 2.9  x
$10^{-3}$ to a maximum 12.3 x $10^{-3}$. As the number of clusters is increased from 2,
the increase in Dunn Index reflects the sequential improvement of the fit as
more clusters are offered to the algorithm to fit the various facets of the data.
In comparison, the Silhouette Width decreases.  Although the similarity of the
elements within each cluster will increase, the dissimilarity between each
cluster will decreases and this what drives the Silhouette Width down.  When
plotted an optimum of 8 clusters was decided upon (average Silhouette Width
of 0.35 and a Dunn Index of 4.6 x $10^{-3}$) based upon these two opposing
factors.  The first factor being the increase in the fit of the clusters to the
natural clusters within the data with increased cluster number and the second
being the over clustering of the data such that the natural clusters are divided
according to the natural spread of the points within the cluster.  This can be
determined by looking for so called 'knees' within the two plots.



**2.3 Meteorological and other data**

**2.4  Air mass trajectories**

Air mass backtrajectories were calculated using the HYSPLIT4 trajectory model (Draxler and Hess, 1998) using the NCAR/NCEP 2.5-deg global reanalysis archive (Kalnay et al., 1996). Trajectories were calculated arriving at Halley (Lat. 75°34'16"S, Long. 25°28'26"W, 30m above sea level (asl)) every 6 hours (06:00, 12:00, 18:00, 00:00) during the study period. All calculations were carried out through the Openair trajectory functions in Cran R (Carslaw and Ropkins 2012). In particular, once calculated, the trajectories were clustered using the Openair function *trajCluster* using the Euclidean method. When considering the various cluster numbers, a setting of 6 trajectory clusters were chosen as best describing the air masses arriving at Halley. Note that metrics similar to the Dunn Index and Silhouette Width were not needed in this decision. The results of the air mass trajectory calculation were plotted either as individual, average or raster layer objects (Hijmans (2019)) drawn on stereographic projections of Antarctica using the *mapproj* and *maps* package (Becker 2018, Doug McIlroy *et al* 2018).

**3. Results**

**3.1 Categorizing Antarctic aerosol size distributions**

**3.1.1  Average particle number and size resolved concentrations**

We investigated the seasonal variability in the physical aerosol size characteristics of particles sampled from Halley VI Station in coastal Antarctica over the period January to December 2015. A clear maximum at 45 nm and at 145 nm can be seen in the annual average size distribution (Fig. 1). However, a striking difference can be seen among different seasons: high concentrations of aerosols at about 40 nm dominate during summer, whereas larger modes can be observed during winter; with intermediate conditions





during spring and autumn. The difference between spring and autumn at
D>60 nm is also interesting, showing much higher concentrations in autumn.
Results are broadly in line with previous results published from the Antarctic
Penininsula (Kim et al., 2017). Total particle number concentrations are
derived from a condensation particle counter (CPC) deployed parallel to the
SMPS (Fig. SI 1), supporting the excellent performance of the SMPS over a
large data coverage (89% of the time during 2015). Minimum concentrations
are found for the month of August ($47\pm10$ cm$^{-3}$) and maximum for January
($602\pm65$ cm$^{-3}$). These are reflected in the clear seasonal cycles for the total
particle concentration (CN) observed (Fig SI 2). Figure SI 2 (bottom) also
shows daily average concentrations of the $N_{30\ nm}$, $N_{30\text{-}100\ nm}$ and $N_{>100\ nm}$
integral particle population. The selected cutoffs of 30 and 100 nm are based
on the average shape of the size distribution (Figure 1). It is interesting that
whereas the absolute concentrations are remarkably different, the relative
percentages of the three aerosol populations do not differ much across
different months, on average $21\pm9\%$, $54\pm7\%$ and $25\pm8\%$ for the $N_{30\ nm}$, $N_{30\text{-}100}$
$_{nm}$ and $N_{>100\ nm}$, respectively. Ultrafine particles dominate summer
concentrations, but are - relative to total - a dominating fraction also during
winter.
**3.1.2 K-means SMPS cluster analysis**
K-means cluster analysis of particle number size distributions was performed
using 5,664 hourly distributions collected over the year of 2015. Our clustering
analysis led to an optimum number of eight categories of aerosol number size
distributions. The corresponding average daily aerosol number size
distributions are shown in Figure 2a, whereas the annual seasonality is shown
in Figure 2b. Here, we refer to ultrafine as particles with diameters between 6
and 210 nm. Three categories were characterized by very low particle number
concentrations (<200 particles cm$^{-3}$), and described by their different aerosol
modes (plotted and size resolved in Fig. 3), specifically:
- "*Pristine_30*" ultrafine. Occurring annually 19% of the time (min-max 0-55%
based on monthly averages), this aerosol category ($N_{CPC}$ $179\pm30$ cm$^{-3}$) shows



two main peaks at 30 nm and 95 nm (Fig. 3, Fig. SI 3). The maximum in
occurence is seen for the months of September (47%) and May (55%).
- "*Pristine_75*" ultrafine. Occurring annually 29% of the time (min-max 0-61%
based on monthly averages), this aerosol category ($N_{CPC}$ 157±25 cm$^{-3}$) shows
two main peaks at 70 nm and 130 nm (Fig. 3, Fig. SI 3). The occurence is
scattered across all year except during spring months (Sept/Oct).
- "*Pristine_160*" ultrafine. Occurring annually 9% of the time (min-max 0-52%
based on monthly averages), this aerosol category ($N_{CPC}$ 121±40 cm$^{-3}$) shows
two main peaks at 70 nm and 160 nm (Fig. 3, Fig. SI 3). The maximum in
occurence is seen for the winter months of June (41%) and July (52%).
These three pristine aerosol cluster types describe up to 57% of the aerosol
population, and mainly dominate the aerosol population during cold months
(73%-100% for Apr-Aug.) Other aerosol categories possessing higher particle
concentrations include:
- "*Nucleation*" ultrafine. Occurring annually 3% of the time (min-max 0-11%
based on monthly averages), this aerosol category ($N_{CPC}$ 620±220 cm$^{-3}$)
shows a main nucleation peak at 15 nm detected during summer months (Fig.
2 a, b). Figure SI3d shows the evolution of the aerosol number size
distributions starting at about noon and peaking at about 18:00; overall 95% of
these events were detected during daylight. The name of this category - which
will be used below to represent new particle formation events - stands for
continuous gas-to-particle growth occurring after the particle nucleation event,
although these nucleation events - detected at about 7-10 nm - must have
orginated away from the Halley station.
- "*Bursting*" ultrafine. Occurring annually 9% of the time (min-max 0-37%
based on monthly averages), this aerosol category ($N_{CPC}$ 602±120 cm$^{-3}$)
shows a main nucleation peak at 27 nm detected during summer months (Fig.
2a, b). Fig. SI3e suggests these aerosols are similar to the *Nucleation* cluster,



although these new particle formation events are already in the growth
process almost reaching 30 nm on average.
Clusters *Nucleation* and *Bursting* are seen during summer months and
September-October, contributing up to 44% of the total aerosol population
during the months of September and January (Fig. SI4b, d). Following
terminology developed in previous work (Dall´Osto et al., 2017, 2018) the
remaining aerosol clusters can be classified as folwed:
- "*Nascent*" ultrafine. This category occurs annually 10% of the time, with a
strong seasonal trend peaking during summer (October-December, 10-39%)
and with a broad Aitken mode centred at about 38 nm (Fig.2) without showing
a clear diurnal pattern (Fig. SI3f). The name of this category emerges from
growing ultrafine aerosol particles which may result from an array of different
primary and secondary aerosol processes.
- "*Aitken*" ultrafine. This category occurs annually 15% of the time, with a
strong seasonal trend peaking during summer (Oct-Dec, 32-63%, Fig. 2b) and
- similar to the *Nascent* cluster - a broad Aitken mode centred at about 50 nm
(Fig 2a) without showing a clear diurnal pattern (Fig. SI 3h).
- "*Bimodal*" ultrafine. Occurring annually 5% (min-max 0-21%) of the time, this
unique category shows a strongly bimodal size distribution (43nm and 134nm,
with a small nucleation mode at 16 nm, Fig. 2 a), it occurs during the period
Dec-Apr (7-21%) and parallels previously reported bimodal aged Antarctic
distributions (Ito et al., 1993). The minimum of the Hoppel mode is seen at 70
nm.
In summary, our method allows apportionment of the Antarctic aerosol
observed at Halley research station into eight categories describing the whole
aerosol population. In the following sections, emphasis is given to
understanding the origin and processes driving Antarctic aerosol formation.





**3.2 Association of PSD with meteorological, physical and chemical**
**parameters**
The main ground-level meteorological observations from Halley for the year
2015 are temporally averaged over the periods of occurrence of the different
aerosol categories (Fig SI 5). Higher average wind speeds (WS, 7.2±2 m s$^{-1}$)
were encountered for the pristine aerosol clusters relative to the remaining
five (3.2±2 m s$^{-1}$); cluster *pristine_160* shows the highest WS (8.5±3 m s$^{-1}$),
suggesting the larger mode may be due to a primary aerosol component,
further discussed in Section 4. Little variation in atmospheric pressure was
found among the eight aerosol clusters. By contrast, *Nucleation* and *Bursting*
clusters were found in driest (Relative Humidity RH, 48±5%) and coldest (T -
17±0.2 ºC) weather among all clusters, supporting the fact that NPF takes
place preferentially at low RH (Laaksonen et al.; 2009; Hamed et al. 2011).
Vertical profiles of meteorological data are available for most days in 2015,
and complement local ground-level measurements. Fig. SI6a-b show driest
and coldest conditions for clusters *Bursting* and *Nucleation*. By contrast,
warmest and wettest conditions occur for the *Bimodal* category.  A large
difference is also seen in the wind speed vertical profiles (Fig. SI 6c), which
are strongest for cluster *pristine_160*, and a clear inversion is seen during the
*bimodal* cluster days. Concurrent ozone gas measurements (Fig. SI 5) show
lowest values for the cluster *bimodal* (18±3 ppb), moderate for ultrafine
dominating clusters (24±8 ppb), and higher values for pristine clusters (29±5
ppb).
**3.3 Elucidating source regions by association of PSD clusters with air**
**mass back trajectories**
Throughout the studied period, hourly 120 h back trajectories were calculated
using the HYSPLIT4 model (Draxler and Hess, 1998). Figure 4 shows the
results of the air mass back trajectories calculated for Halley throughout 2015,
showing six main clusters. Broadly, two air trajectory clusters were associated
with anticyclonic conditions (clusters 2 and 6, up to 33.6% of air masses);
three clusters were associated with air masses coming from the East Antarctic





Plateau (clusters 3, 4, 5, up to 57.2% of air masses); and one unique air
trajectory cluster was found associated with air masses originating within the
Weddell Sea (cluster 1, 9%). Fig. SI7 shows the six air mass back trajectory
clusters and the average height of the trajectories up to 120 hours before
arrival at Halley. While clusters 2-6 show their origin over the Antarctic plateau,
cluster 1 shows average altitudes lower than 1000m, close to the height of the
mixed layer (Fig. SI 7). On the basis of Figure SI7, it looks rather similar to the
other air mass types with the air only entering the boundary layer for the last
~15 hours of the trajectory. One striking difference is found when these air
mass back trajectory clusters are compared temporally among the aerosol
categories (Figure 5).
A key conclusion of this study is that most aerosol categories (excluding
cluster *Nucleation*) are associated with air masses arriving with Eastern winds
from the Antarctic plateau (East short, East long, 56-76% of the time).
Anticyclones also seem to be a predominant air mass type (17-42%). At
Halley, air mass back trajectories that have travelled over the sea/sea ice
zone, play only a minor overall role in terms of annual average air mass
trajectories (10-15%). In a further analysis, we obtained information on how
far each air mass travelled (total travel time 60 h) over zones distinguished by
their surface characteristics, namely snow, sea ice and open water for each
one of the different aerosol categories presented (see methods). Fig. 5a
shows that category *Nucleation* is the one most associated with sea ice (27%
of the time). It is important to stress that the *Nucleation* category has its air
mass back trajectories mainly travelling over land (63%). However - relative to
the other clusters - it is the most affected by air masses which had travelled
over the Weddell Sea (27%), most of which is open pack ice (ratio open pack /
consolidated sea ice of 0.6, Fig. 5b). This is an important conclusion of this
work, pointing out that at least two source regions of new particle formation
exist in the Antarctic. It is interesting to note also that the *Bursting* category
has a large ratio of open pack / consolidated sea ice (Fig 5b), confirming
marginal sea ice zones may be a strong source of biogenic gases responsible
for new particle formation.
By examining the air mass trajectory heights, we also show that during the 5
days prior to sampling, the sampled air from the Weddell Sea was remarkably



different from the other air mass types (Fig. SI 7); it had travelled within the marine boundary layer, with no intrusion from the free troposphere. Our results strongly suggest the nucleating events originated within the boundary layer, likely from gaseous precursors associated with sea ice emissions.

## 4. Discussion

### 4.1 Origin and sources of Antarctic aerosol

The purpose of this study was to analyze a year-long (throughout 2015) set of observations of Antarctic aerosol number size distributions to gain a better understanding of those processes which control Antarctic aerosol properties. In a pristine environment like Antarctica and its surrounding ocean, where the atmosphere is thought to still resemble that of preindustrial Earth (Hamilton et al., 2014), missing aerosol sources must reflect overlooked natural processes. Uncertainties for modeling aerosol-cloud interactions and cloud radiative forcing arise from a poor source apportionment of aerosols and their size distributions (Carslaw et al 2013).

Broadly, marine particles in the nanometer size range originate from gas-to-particle secondary processes, whereas those in super-micron sizes are predominantly composed of primary sea-spray (O´Dowd et al., 1997). However, the accumulation mode (broadly composed of intermediate particle sizes of 50 –500 nm) is composed of a complex mixture of both secondary and primary particles. The relative roles of secondary aerosols produced from biogenic sulfur versus primary sea-spray aerosols in regulating cloud properties and amounts above the Southern Ocean is still a matter of debate (Meskhidze and Nenes, 2006; Korhonen et al., 2008; Quinn and Bates, 2011; Mc Coy et al., 2015; Gras and Keywood, 2017; Fossum et al., 2018). First observations of organic carbon (OC) in size-segregated aerosol samples collected at a coastal site in the Weddell Sea (Virkkula et al., 2006) showed that MSA represented only a few % of the total OC in the submicron fraction; recent studies demonstrate that sea bird colonies are also important sources



of organic compounds locally (Schmale et al., 2013; Liu et al., 2018) and from seasonal ice microbiota (Dall'Osto et al., 2017). The overall balance between secondary aerosol formation versus primary particle formation from sea spray still needs to be determined and is a pressing open question.

A key result of this study is that for 59% of the year (89-100% during winter JJA; 10-50% during spring SON; 34-65% during summer DJF; 48-91% during autumn MAM), aerosol size distributions were characterized by very low particle number concentrations (< 121-179 cm$^{-3}$). It is often assumed that a strong annual cycle of particle number concentrations is mainly driven by summer new particle formation events (Shaw, 1988; Ito et al., 1993; Kerminen et al., 2018). However, at Halley during summer 2015, 34-65% of the time low particle number concentrations of unknown origin dominate the overall temporal variation.  Unique bimodal size distributions are seen in December-April, where a clear bimodal distribution is seen for 7-21% of the time (peaking in March, 21%), and likely related to cloud processing (Hoppel et al., 1994).

In the following sub-sections we discuss our results in the light of recent studies focusing on Antarctic aerosol source apportionment. The majority of the studies report primary and secondary components in term of mass, which should not be confused with particle number concentration.

### 4.1.1 Primary Antarctic aerosol

Sea spray is almost always reported as the main source of supermicron (>1 µm) aerosols in marine areas, including the Southern Ocean and Antarctica (Quinn et al., 2015; Bertram et al., 2018). However, models of global sea-salt distribution have frequently underestimated concentrations at polar locations (Gong et al., 2002). Rankin and Wolff (2003) suggested the Antarctic sea ice zone was a more important source of sea salt aerosol, during the winter months, than the open ocean. In particular, they proposed brine and frost flowers on the surface of newly forming sea ice as the dominant source, a hypothesis supported by other studies (e.g. Udisti et al., 2012). The results presented here suggest that, in coastal Antarctica, aerosol composition is a strong function of wind speed and that the mechanisms determining aerosol





composition are likely linked to blowing snow (Giordano et al., 2019; Yang et
al., 2019; Frey et al., 2019). We note that Legrand et al. (2017a) suggested
that on average, the sea-ice and open-ocean emissions equally contribute to
sea-salt aerosol load of the inland Antarctic atmosphere.
Averaged across the year, we found a very clear aerosol size distribution with
the largest detected mode at ~160 nm, pointing to a primary - likely sea spray
- source, which was detected during periods of strong winds. However, it is
also possible that in size range the dominating constituent is sulphate (Teinilä
et al., 2014), further studies are needed to apportion this mode correctly. This
aerosol category type occurs very frequently during winter months (JJ, 33-
52%), but not during the other months (0-14%). Gras and Keywood (2017)
showed, using data from Cape Grim, that wind-generated coarse-mode sea
salt is an important CCN component year round and from autumn through to
mid-spring is the second most important component, contributing around 36%
to observed CCN; these measurements were taken in the Southern Ocean
marine boundary layer.
Marine primary organic aerosol (POA) is often associated with sea-spray, but
recent studies indicate that a fine mode (usually <200 nm) can have a size
distribution that is independent from sea-salt (externally mixed), whereas
supermicron marine aerosols are more likely to be internally mixed with sea-
salt (Gantt and Meskhidze, 2013). McCoy et al. (2015) reported observational
data indicating a significant spatial correlation between regions of elevated
Chl-a and particle number concentrations across the Southern Ocean, and
showed that modeled organic mass fraction and sulphate explains 53 ± 22%
of the spatial variability in observed particle concentration. Our study cannot
apportion any aerosol related to primary organic aerosol, given the lack of
chemical measurements carried out during 2015 at Halley research station. It
is possible that part of the broad mode at 90 nm of the Pristine_90 category
contain a fraction of primary marine organic aerosols, but the relative
importance cannot be quantified in this study. Interestingly, open ocean
aerosol measurements collected over the Southern Ocean (43°S–70°S) and
the Amundsen Sea (70°S–75°S) were recently reported by Jung et al. (2019).
During the cruise, Water Insoluble Organic Components (WIOC) was the





dominant Organic Carbon (OC) species in both the Southern Ocean and the
Amundsen Sea, accounting for 75% and 73% of total aerosol organic carbon,
respectively. The WIOC concentrations were found to correlate with the
relative biomass of a specific phytoplankton species (P. Antarctica), producing
extracellular polysaccharide mucus and strongly affecting the atmospheric
WIOC concentration in the Amundsen Sea (Jung et al., 2019).
**4.1.2 Secondary Antarctic aerosol**
Our results show that two sub 30 nm aerosol categories (*Nucleation* and
*Bursting*, 12% in total) and two Aitken 30-60 nm aerosol categories (*Nascent*
and *Aitken*, 25%) account for up to 37% of the PSD detected during at Halley
the year 2015. Our results point to secondary aerosol processes driving the
aerosol population during five months of the year (Sep-Jan, 48-90%), where
aerosol particle number concentrations are on average 3-4 higher than the
Antarctic aerosol baseline. Our study strongly suggests that new particle
formation may have at least two contrasting sources. The former is related to
sea ice marginal zones formed in the marine boundary layer. The latter is
related to air masses arriving from the Antarctic plateau, possibly having a
free troposphere origin.
The biogenic precursors responsible for the new particle formation are not
known. Charlson et al. (1987) postulated the CLAW hypothesis - the most
significant source of CCN in the marine environment is non-sea-salt sulfate
derived from atmospheric oxidation of dimethylsulfide (DMS); however
measurements able to provide information on where individual particles come
from are still limited (O´Dowd et al., 1997b; Quinn and Bates, 2011; Sanchez
et al., 2018). A previous ship-borne field campaign in the Weddell Sea found
increased new particle formation in the sea ice zone of the Weddell Sea
(Davison et al., 1996), but no clear correlation to the dimethyl sulphide that
was then assumed to control new particle bursts. A smaller mode radius
associated with polar aerosol (relative to marine Southern ocean aerosol) was
found associated with less cloud cover, and consequently less cloud
processing, over the continent and pack ice regions. During the cruise, new
particle formation observed over the Weddell Sea, resulted from boundary



layer nucleation bursts rather than tropospheric entrainment. Brooks and
Thornton (2018) argued that additional modeling studies are still needed that
address contributions from both secondary DMS-derived aerosols and primary
organic aerosols as CCNs on realistic timescales; although the occurrence of
a "seasonal CLAW" in remote marine atmospheres is becoming plausible
(Vallina and Simó, 2007; Quinn et al., 2017; Sanchez et al., 2018).
Satellite (Schonhardt et al., 2008) and on-site measurements (Saiz-Lopez et
al., 2007; Atkinson et al., 2012) showed that the Weddell Sea is an iodine
hotspot; however there was no short-term correlation between IO and particle
concentration found (Roscoe et al., 2015). Using an unprecedented suite of
instruments, Jokinen et al. (2018) showed that ion-induced nucleation of
sulfuric acid and ammonia, followed by sulfuric acid–driven growth, is the
predominant mechanism for NPF and growth in eastern Antarctica a few
hundred kilometers from the coast (Finnish Antarctic research station (Aboa)
is located at the Queen Maud land, Eastern Antarctica; Jokinen et al., 2018).
Some ion clusters contained iodic acid, but its concentration was very small,
and no pure iodic acid or iodine oxide clusters were detected (Sipila et al.,
2016). Finally, some organic oxidation products from land melt ponds have
also been suggested (Kyro et al., 2013) as a potential source for condensable
vapor, although this may be a confined and minor source (Weller et al., 2018).
Other measurements of new particle formation and growth were governed by
the availability of other yet unidentified gaseous precursors, most probably low
volatile organic compounds of marine origin (Weller et al., 2015; 2018).

## 4.2 Implication for climate and conclusion

A strong annual cycle of total particle number concentration is a prominent
characteristic of the Antarctic aerosol system, with the austral summer
concentration being up to 20-100 times greater than during the winter (Shaw
1988, Gras 1993, Ito 1993, Hara et al 2011, Weller et al 2011, Järvinen et al
2013, Fiebig et al 2014, Kim et al 2017). These summer particle number
concentration maxima are largely explained by NPF taking place in the





Antarctic atmosphere. However, these seasonal cycles are more pronounced
at monitoring sites situated on the upper plateau of Antarctica than at the
coastal Antarctic sites. It is worth to keep in mind that these cycles could also
be more pronounced because in coastal regions in winter, sea salt aerosol
has a relatively larger source. i.e. the amplitude of the seasonal is driven both
by what is going on in winter as well as summer. Nevertheless, overall much
higher particle number concentrations have long been reported in coastal
Antarctica relative to the plateau. The vertical location of Antarctic NPF has
not been well quantified; there are some indications that NPF takes place
preferentially in the Antarctic Free Troposphere (FT) rather than in the
Boundary Layer (BL) (Koponen et al 2002, Hara et al 2011, Humphries et al
2016), whereas other studies shows opposite trends (Kim et al., 2017, Weller
et al., 2011; 2013; 2018). A study conducted on the upper plateau of
Antarctica demonstrates that also wintertime regional NPF is possible in this
environment (Järvinen et al 2013). Very low particle growth rates (between
about 0.1 and 1 nm h$^{-1}$) were reported in Antarctica (Park et al 2004, Weller et
al 2015).
We obtained data from Dome C and King Sejong (KS) Station for the period
May-December 2015, and compared them with Halley (H). Data are shown in
Fig. 6 where seasonal mean aerosol size distributions measured
simultaneously at three different sites are reported for (a) May-December
2015 (8 months in total); (b) Spring (September, October, November, 3
months in total); (c) Summer (December, 1 month in total) and (d) Winter
(June, July, August, 3 months in total, a map of the three stations considered
is shown in Figure 7. Overall, much higher concentrations are seen at the
coastal Antarctic sites (H, KS stations) relative to Dome C station (Fig. 6a).
Two broad modes at about 30-50 nm and at about 110-160 nm can be seen
for the coastal stations, whereas a smaller single mode at 60 nm is seen for
the Dome C station. When three seasons are compared, very different
features can be seen. During spring (Fig. 6b), both Aitken and accumulation
modes dominate the coastal sites, whereas a strong single mode is seen in
the Dome C site. By contrast, during summer (Fig. 6c), much stronger
nucleation and Aitken modes are seen at the coastal sites, likely due to NPF



taking place during summer time. The smaller nucleation mode size detected
in the Antarctic peninsula (King Sejong Station) relative to the one seen at
Halley may suggest a more local source of NPF in the Antarctic peninsula,
including open water, coastal macroalgae, and bird colonies. The average
size distributions during winter (Fig. 6d) again show marked differences
among the three different monitoring sites. Halley stations shows the largest
aerosol modes (about 100 nm and 160 nm), whereas smaller modes can be
seen at the other two sites. Overall, Fig. 6 serves to stress that the aerosol
population in Antarctica - an environment often considered homogenous and
simple to study - is different in different geographical regions, and very likely a
number of different processes and sources affect the aerosol population at
different times of the year. Ito et al. (1993) presented a conceptual diagram,
where different aerosol size distributions were seen, and a main NPF mode
was associated with the free troposphere and transported by katabatic winds.
Korhonen et al. (2008) also estimated that over 90% of the non-sea spray
CCN were generated above the boundary layer by nucleation of sulfuric acid
aerosol in the free troposphere.  Our results point to sea ice regions and open
ocean water being a source not only of gaseous precursors, but also of new
particle formation, which then can growth once lifted in the free troposphere
(Fig. 8), and then larger modes are brought down again by the Antarctic
Drainage flow (James, 1989). The relative importance of free troposphere
versus boundary layer nucleation is not known at this stage, but this study
shows that the latter is seen, and the former is likely to happen and contribute
to the Aitken mode detected from the Antarctic plateau. Sea ice regions
(mainly via secondary processes, but also to a lesser degree via sea spray
and blowing snow) may control the CCN production, both regulating the first
stage of nucleation events and providing gaseous precursors, and slowly
growing nucleated particles with transport in the upper troposphere.
These results are in line with previous studies in polar areas. First, Dall´Osto
et al (2017) suggested that the microbiota of sea ice and sea ice-influenced
ocean were a significant source of atmospheric nucleating particles
concentrations ($N_{1-3nm}$). Second, within two different Arctic locations, across
large temporal scales (2000-2016) new particle formation was associated with



air mass back trajectories passing over open water and melting sea ice regions, also pointing to marine biological activities within the open leads in the pack ice and/or along the melting marginal sea ice zone (MIZ) being responsible for such events (Dall´Osto et al., 2017b, Dall´Osto et al., 2018). Our data from Halley, and the brief intercomparison with two other stations, suggest that the size distributions of Antarctic submicron aerosols may have been oversimplified in the past (Ito et al., 1993); and complex interactions between multiple ecosystems, coupled with different atmospheric circulation, result in very different aerosol size distributions populating the Southern Hemisphere.

**Acknowledgements**

The authors are grateful to the overwintering staff at Halley station who carried out the suite of measurements presented here. This work was funded by the Natural Environment Research Council as part of the British Antarctic Survey's research programme "Polar Science for Planet Earth". The study was further supported by the Spanish Ministry of Economy through project PI-ICE (CTM 2017–89117-R) and the Ramon y Cajal fellowship (RYC-2012-11922). The National Centre for Atmospheric Science NCAS Birmingham group is funded by the UK Natural Environment Research Council. We thank Dr. Pasi aalto (Institute for Atmospheric and Earth System Research, University of Helsinki),  for providing DMPSdata of 2015 for intercomparison with data taken at Halley Station, similar data were discussed in details elsewhere (Järvinen  et al., 2013; Kim et al., 2017) . AV as supported by the Academy of Finland's Centre of Excellence program (Centre of Excellence in Atmospheric Science – From Molecular and Biological processes to The Global Climate, project no. 272041).  KS station SMPS measurement was supported by KOPRI project (PE19010).





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

13 standard deviation of the measurements from the mean value.



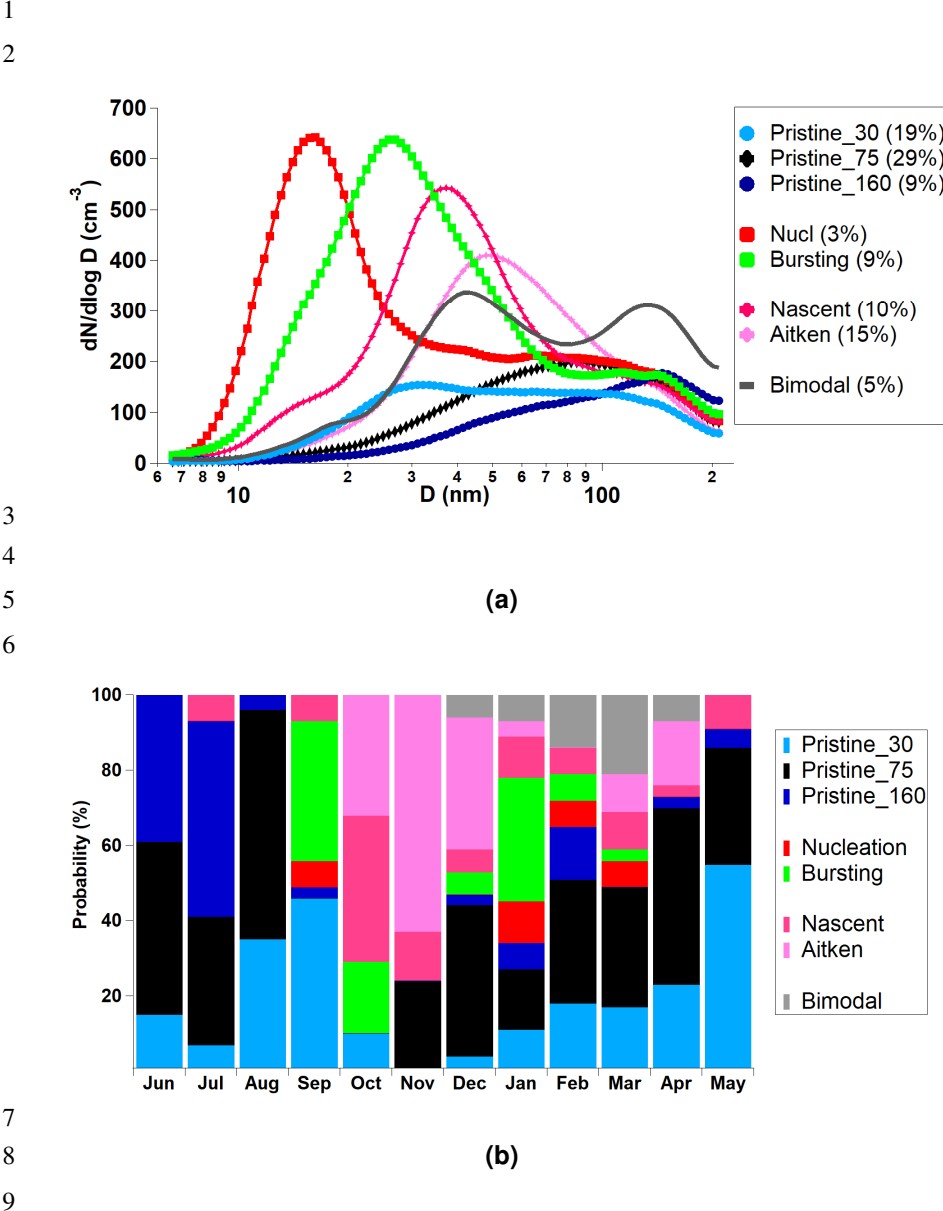

**(a)**

**(b)**

**Figure 2**   (a) Size distributions of the 8 k-means clusters and (b) annual frequency distributions of the six aerosol categories








**Figure 3** Peak fitting of the 3 pristine K-means aerosol categories.

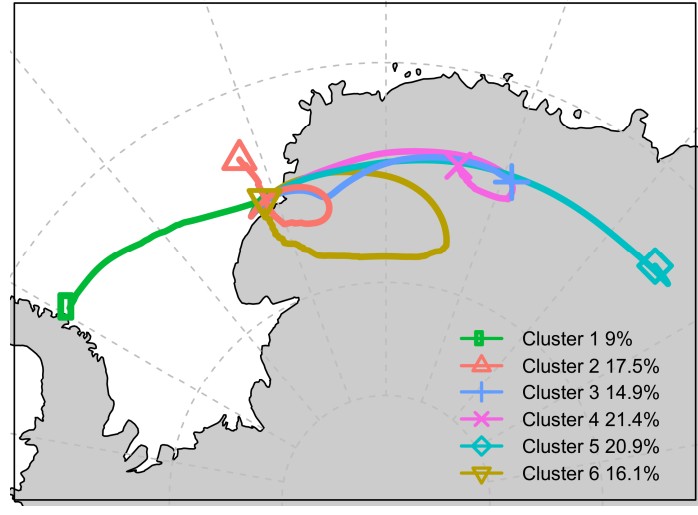

**(a)**

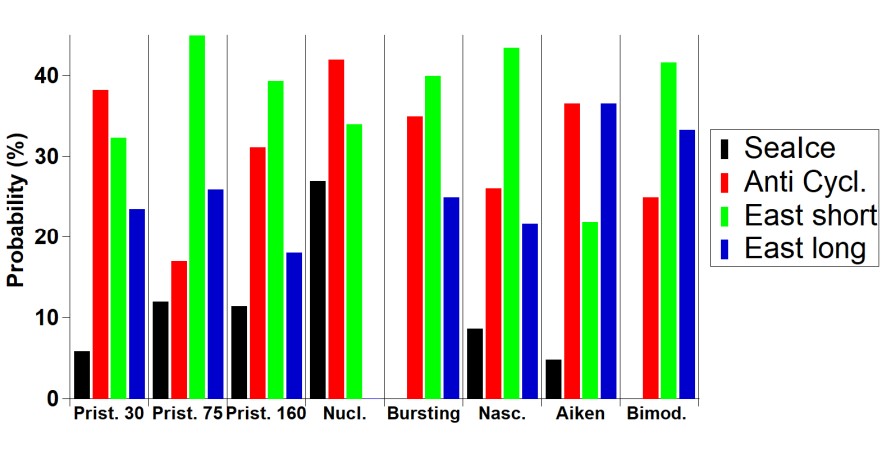

**(b)**

**Figure 4 (a**) Air mass analysis of air mass back trajectories arriving at Halley
during the year 365 (hourly resolution) and **(b)** relative contribution for each
aerosol category. Groups in (b) are : Sea Ice (1), Anti Cycl (2,6), East short
(3,4) and east long (5),





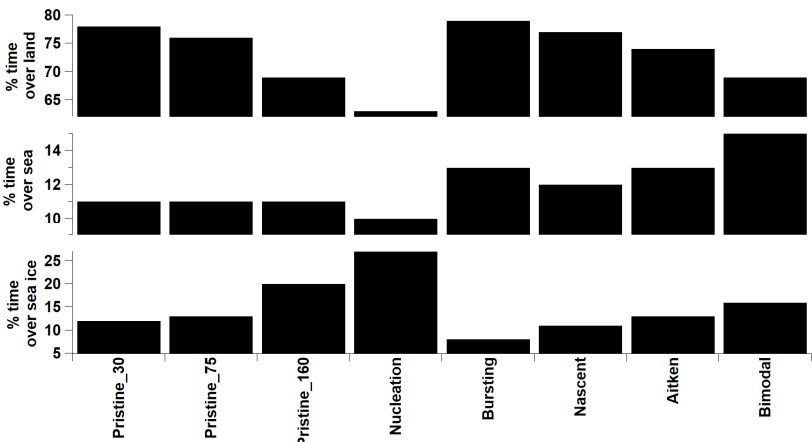

**(a)**

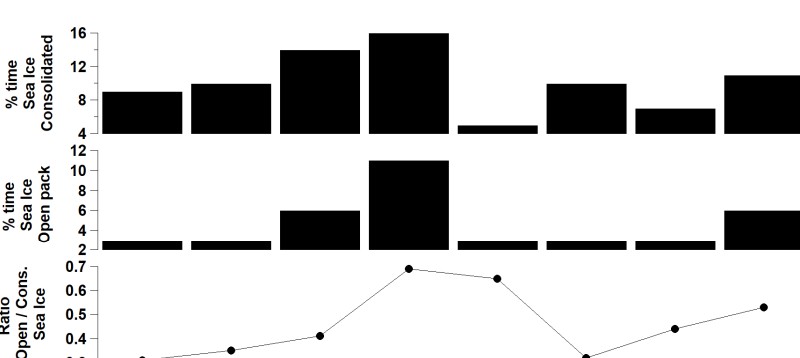

**(b)**
**Figure 5** (a) Percentages of air masses over land, sea, and sea ice for the 8
K-means aerosol categories and (b) percentages of consolidated and open
pack sea ice, and open pack / consolidated ratio.





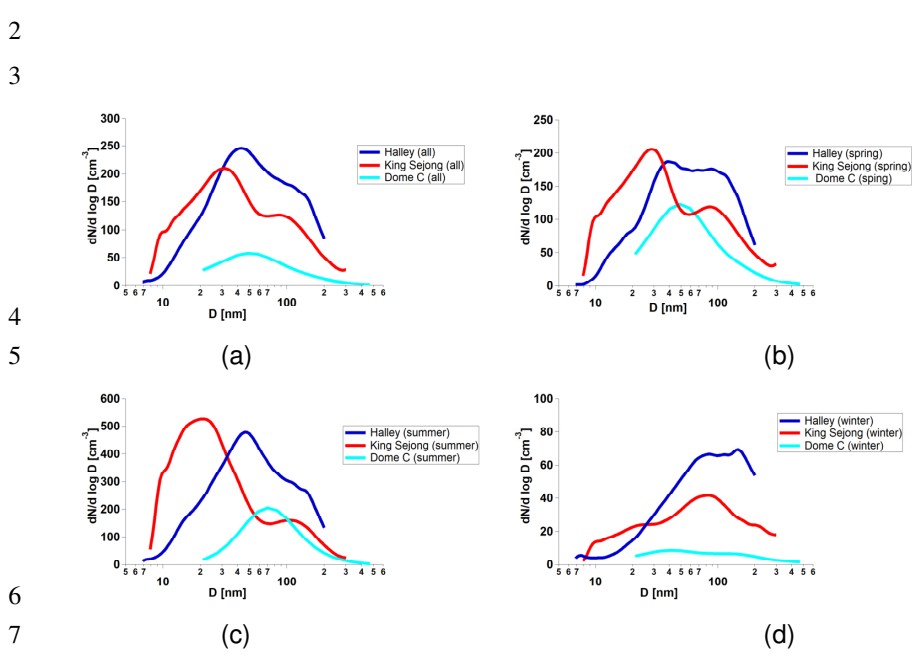

5          (a)                                               (b)

7          (c)                                               (d)

**Figure 6.** Average size-resolved particle size distributions simultaneously
measured during the year 2015 at Halley, Dome C and King Sejong stations
for (a) May-December (8 months), (b) spring (Sep., Oct., Nov., 3 months), (c)
summer (December, 1 month) and (d) winter (Jun., Jul., Aug., 3 months).





**Figure 7.** Map with locations of Antarctic monitoring stations considered in Figure 6. Please note that the sea ice extent is the median September extent from 1981-2010 (data are from NSIDC - https://nsidc.org/data/g02135).



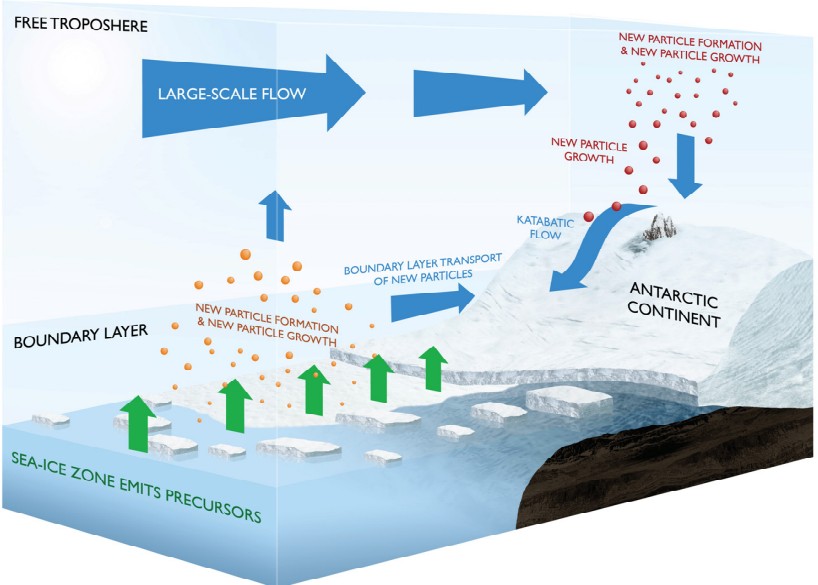

**Figure 8** Schematic illustrations of the ultrafine New Particle Formation (NPF)
and New Particle Growth (NPG) aerosols in Antarctica.
