# Peer review of "On the annual variability of 2 Antarctic aerosol size 3 distributions at Halley research 4 station 5 6 Thomas Lachlan-Cope1, David Beddows2, Neil Brough1, Anna E. 7 Jones1, Roy M. Harrison2,+, Angelo Lupi3, Young Jun Yoon4</s"

_Atmospheric Chemistry and Physics, 2019_

## Referee Comment (RC1) · Anonymous Referee #1 · 21 Oct 2019

The authors analyzed a unique data set on year-round particle size distribution (PSD) measured at the coastal Antarctic station Halley. They based their data evaluation on statistical cluster analysis, which has been applied as beneficial tool in several comparable investigations (References: Dall'Osto et al., 2019, 2018, and 2017). The manuscript at hand presents valuable, meaningful, and novel findings from a region where only very few studies on the variability of aerosol physical properties are available. Without doubt, the topic addresses the scientific scope of ACP, particularly considering the fact that aerosol-cloud interaction in the southern Ocean realm is still poorly understood leading to strong biases in climate modelling. Most notably in this context, PSD measurements from this region are qualified for assessing the potential of the aerosol to act as cloud condensation nuclei (CCN). Hence, I recommend a final

publication after some more or less basic revisions.

General issues:

(i) Presentation and discussion of the results are largely restricted to the "higher-level" output of the cluster calculations. Therefore, you should clearly substantiate the advantages and benefits of this method. The short section provided on page 6, lines 1 to 8 appears scarce. To be more specific (or even provocative): Two of the main conclusions drawn from this study and mentioned in the Abstract as point (1) and (2) (page 2, lines 22 to 28) can be easily derived without using any cluster analysis.

(ii) Moreover, from my point of view, it would be beneficial or even necessary to focus from case to case more on the original SMPS data, primarily when assigning air mass origins to NPF events. Here a more detailed discussion of air mass histories along with the original, individual PSD-spectra could be much more meaningful (the sketchily approach presented on page 14, lines 12 to 32 is hardly adequate). In case of "Nucleation" cluster: Do the individual PSD-spectra show particle growth in contrast to the spectra assigned to "Bursting"? Especially here, you may present some examples from the original data set to demonstrate the unique characteristic.

(iii) Air mass back trajectory analysis is a fundamental scaffolding of this study. The trajectory cluster analysis is interesting on its own but, however, somewhat detached from the PSD cluster analysis. I recommend presenting a figure analogous to Fig. SI 7 in the main text, but showing here trajectory ensembles sorted according to the PSD clusters as described on page 14, lines 12 to 32. Just another (minor) point concerning Fig. SI 7: The plot for cluster 1 (sea ice) shows terrain heights typically around 200 m or so, though the air masses travelled across the Weddell Sea (terrain height should be around zero!) – please check and clarify!

(iv) I recommend moving Figures SI 3 and SI 4 presented in the Supplementary Information (SI) to the main text, because they contain crucial information.

(v) Whenever possible, provide corresponding uncertainties or standard deviations of the results, especially for any values given in "%" (regarding text and figures).

(vi) The pivotal question you raise addresses the balance between secondary vs. primary aerosol in this region (see Abstract lines 8 to 11 and p. 16, lines 2 to 4). I suggest picking up this quest in your conclusions more explicitly. Finally: Do you have any suggestion for future research on this topic?

Some specific and minor points:

1. Abstract: Please concretely state here size range, temporal resolution and measuring period.

2. Page 4, line 29: . . .higher NPF instead of higher N.

3. Chapter 3.2: The association of PSD with meteorology, physical and chemical parameters appears rather descriptive. Do you have any ideas regarding the physical background of your findings?

4. Page 9, lines 17 to 19: Hijman (2019) and Becker (2018) are not listed in the references.

5. Page 14, line 19: Why did you relate to a total travel time of just 60 h and not 120 h (5 days back trajectories)?

6. Page 16, line 12: Please state "low particle number concentrations" more precisely.

7. Page 17, lines 5 to 16: I guess, during winter nss-sulphate aerosol should be negligible compared to sea salt. Maybe an additional closer look into the material presented in Rankin and Wolff (2003) or previous results on the chemical composition of the bulk aerosol from that site could be revealing, especially assessing the role of primary aerosol acting as CCN.

8. Page 18, line 16: Please state "baseline" more precisely.

[Figure]

9. Figure 4, caption, line 9: ... during the year 2015 (not: during the year 365).

---

## Referee Comment (RC2) · Anonymous Referee #2 · 6 Nov 2019

The authors analyze particle size distribution data from coastal Antarctica using statistical methods to draw conclusions about aerosol sources and atmospheric processes. The results presented are both valuable and novel and are definitely within the scope of ACP. The context of the analysis and some of the actual discussion, especially as it relates to the existing literature, needs to be expanded but the necessary additions are minor with regards to the overall manuscript. Therefore, I recommend final publication with (generally) minor revisions.

Major comments:

Clustering analysis (especially S.2.2.1 and S.3.1.2) – The discussion on the clustering analysis needs to be greatly expanded as this is a fairly novel technique in atmospheric science. Many people fall into the trap of thinking that this machine learning method

is actually machine intelligence and simply gives a "correct" answer as opposed to a mathematically valid solution. First, the values given for the Dunn Index and Silhouette Width need to be given context. Primarily, plots of both versus cluster number should be offered as many readers have no experience using or analyzing cluster analysis results. Secondarily, the values themselves need to be discussed in much greater detail. The 4x increase of the Dunn Index is good but $10^{-3}$ is still an extremely small value and implies that the clusters are extremely sparse (not compact), are not very far apart, or both. A graph of the cluster points to visually inspect both compactness and distance between clusters may be useful but may also be misleading as ambient data sets are often quite messy. Second, some additional validation of the cluster choice must also be presented. One way to do this may be to perform the analysis on heavily curated data to see if the results broadly match the overall analysis. I would highly (and very strongly) recommend that the authors run the analysis on a time frame where there are a minimum number of clusters expected (e.g. June). If there is broad agreement between the results there and the overall results, this would lend a great amount of strength to the overall conclusions. This could, and possibly should, be done in the context of air mass back trajectories as well where air masses could be broadly classified relative to their time spent over the continent, sea ice, or open ocean (this also comes with major caveats though, also see minor comment about back trajectory analysis).

S.2.3 – is missing?

S.4.1 – Much of the length of this section could be moved into the introduction and the remaining text expanded to give a more complete view of how these results fit into the existing literature. Overall, the authors do a fine job of finding relevant papers but do not necessarily discuss the conclusions presented completely. In particular, more discussion regarding measured composition and size distributions and the results presented here may be useful. The results of Rankin and Wolff (2003), Preunkert at al. (2007, 2008), Saiz-Lopez et al. (2007), Schmale et al. (2013), Giordano et al. (2017),

and many others should likely be discussed in greater detail. Additionally, the presence and lack of photochemistry should be given some context as this is a fairly dominating factor in the polar regions' winter vs. summer months.

Diurnal profiles (Fig. SI 3 especially) – The basis of this analysis, especially considering the weight the figure is given in the text itself, needs to be better justified. Diurnal profiles are generally helpful in visualizing the impacts of either photochemistry or timed anthropogenic activities (or both). Neither of these cases apply to the Antarctic continent. Either the analysis should be rerun in a more nuanced approach (e.g. diurnals for periods of 24-hours of sunlight and lack thereof, only run in the short timeframes of clearly demarcated sunrise/sunset) and discussed in that context or should be removed completely. These results could be analyzed to give important insights into the potential role that the Polar sunrise/sunset plays in aerosol size distributions but this analysis may be beyond the scope of this manuscript.

Minor comments:

P.7, S2.2 – A few sentences about the transmission efficiency of the aerosol stack for relevant sizes of aerosols should be added. The authors could consider applying a correction to the size distributions to account for inlet losses but I imagine they are fairly small for the relevant sizes.

S.2.4 and 3.3 – A few sentences regarding the accuracy of HYSPLIT being used in regions of sparse meteorological measurements should be added. A more detailed description of the initialization conditions for the model should also be added.

S.3.3 – The conclusions discussed in this section should be moved to S.4 and discussed in the context of the existing literature.

S.4.2 – The conclusions should be separated from the discussion. The work presented here is worthwhile and the main points should not be hidden.

Overall – consistency in figure references, especially for SI figures, should be double

checked. E.g. Fig. SI 3 @ P.11 L11 vs Fig. SI3e @ P.11 L33).

Overall – references need to be double checked in both the main text and the references section.

References: Rankin, A. M. and Wolff, E. W.: A year-long record of size-segregated aerosol composition at Halley, Antarctica, J. Geo-phys. Res., 108, D244775, doi:10.1029/2003JD003993,2003.

Preunkert, S., Legrand, M., Jourdain, B., Moulin, C., Belviso, S.,Kasamatsu, N., Fukuchi, M., and Hirawake, T.: Interannual vari ability of dimethylsulfide in air and seawater and its atmosphericoxidation by-products (methanesulfonate and sulfate) at Dumontd'Urville, coastal Antarctica (1999–2003), J. Geophys. Res.,112, D06306, doi:10.1029/2006JD0075857, 2007.

Preunkert, S., Jourdain, B., Legrand, M., Udisti, R., Becagli,S., and Cerri, O.: Seasonality of sulfur species (dimethyl sul-fide, sulfate, and methanesulfonate) in Antarctica: Inland ver-sus coastal regions, J. Geophys. Res. Atmos., 113, D15302,doi:10.1029/2008JD009937, 2008.

Saiz-Lopez, A., Mahajan, A. S., Salmon, R. A., Bauguitte, S. J. B.,Jones, A. E., Roscoe, H. K., and Plane, J. M. C.: Boundary LayerHalogens in Coastal Antarctica, Science, 317, 348–351, 2007.

Schmale, J., Schneider, J., Nemitz, E., Tang, Y. S., Dragosits, U.,Blackall, T. D., Trathan, P. N., Phillips, G. J., Sutton, M., andBraban, C. F.: Sub-Antarctic marine aerosol: dominant contri-butions from biogenic sources, Atmos. Chem. Phys., 13, 8669–8694, doi:10.5194/acp-13-8669-2013, 2013.

Giordano, M. R., Kalnajs, L. E., Avery, A., Goetz, J. D., Davis, S. M., and DeCarlo, P. F.: A missing source of aerosols in Antarctica – beyond long-range transport, phytoplank-ton, and photochemistry, Atmos. Chem. Phys., 17, 1–20, https://doi.org/10.5194/acp-17-1-2017, 2017.

---

## Referee Comment (RC3) · Anonymous Referee #3 · 11 Nov 2019

Comments: Lachlan-Cope et al. present a novel study of Antarctic aerosol size distribution collected over a whole year of measurements at Halley research station. By applying the K-means clustering data analysis, eight aerosol categories were characterized. Based on the air mass back trajectory analysis, major sources regions including sea ice, open ocean, snow, and etc. were elucidated. Then, this study concluded that NPF and growth events in the Antarctic atmosphere mainly originated from both the sea ice marginal zone and the Antarctic plateau. The implication for climate and conclusion section is well-written, and in particular, the brief comparison with two other Antarctic stations (Dom C Concordia and King Sejong Station) during the year 2015 is a very useful and insightful section. Overall, the manuscript is generally well-written and interesting to read, with clear structure and sufficient explanations. The manuscript

may be suitable to be published in Atmospheric Chemistry and Physics.

Major Comments: Page 2 and Lines 30: In the present study, cluster "pristine_160" with a bimodal size distribution (75 nm and 160 nm, respectively) shows the highest WS, but there were no correlations between them. Please clarify the meaning of "strong correlation in the abstract". Page 11 and Lines 19: New particle formation and growth was observed for the nucleation mode PSC cluster. In addition, the authors mentioned that NPF and growth events originate from both sea ice marginal zone and the Antarctic plateau in the abstract. Please, calculate and suggest the growth rate, which is a critical factor that affects the CCN number concentration in Antarctic regions. Then, the value could be compared according to the air mass origins. Page 15 and line 9. Almost 5 pages were partitioned to discussion section that was overlapped with introduction section. Most of of the discussion section should be moved to introduction section and SI. Page 20 and line 19. The authors compared data from Halley, Dome C, and King Sejong Station. Overall, much higher concentrations are seen at the coastal Antarctic sites relative to continental based Dome C station. The coastal Antarctic stations being a remote location might be not immune to man-made impacts and specific tracers (e.g., black carbon) are necessary to discern those influence. In particular, quite high BC concentration was detected in King Sejong Station, as presented by Kim et al. (2018). Here, the possible sources of NPF and growth due to human activity (anthropogenic influence) could be discussed.

Minor Comments: Pate 9 and Line 1: Section 2.3 is missing. Page 10 and Line 1: As mentioned in the manuscript, the difference between spring and autumn at Dp > 60 nm is very interesting. Please, explain possible reasons. Page 2 and Lines 30: Error should be displayed in Figure SI 5. Please, provide the relationships between total particle number concentration and each meteoroidal data (e.g., wind speeds, RH, T, and ozone) according to the different aerosol categories. Page 38 and line 6: What is meant by "Please note that the sea ice extent is the median September extent from 1981-2010" in Figure 7.

Please also note the supplement to this comment:
https://www.atmos-chem-phys-discuss.net/acp-2019-847/acp-2019-847-RC3-supplement.pdf
* * *

---

## Author Comment (AC1) · 17 Feb 2020

The authors analyzed a unique data set on year-round particle size distribution (PSD) measured at the coastal Antarctic station Halley. They based their data evaluation on statistical cluster analysis, which has been applied as beneficial tool in several comparable investigations (References: Dall'Osto et al., 2019, 2018, and 2017). The manuscript at hand presents valuable, meaningful, and novel findings from a region where only very few studies on the variability of aerosol physical properties are available.

*Many thanks for appreciating the manuscript.*

Without doubt, the topic addresses the scientific scope of ACP, particularly considering the fact that aerosol-cloud interaction in the southern Ocean realm is still poorly understood leading to strong biases in climate modelling. Most notably in this context, PSD measurements from this region are qualified for assessing the potential of the aerosol to act as cloud condensation nuclei (CCN). Hence, I recommend a final publication after some more or less basic revisions.

*Many thanks again for appreciating the manuscript.*

General issues:

(i) Presentation and discussion of the results are largely restricted to the "higher-level" output of the cluster calculations. Therefore, you should clearly substantiate the advantages and benefits of this method. The short section provided on page 6, lines 1 to 8 appears scarce. To be more specific (or even provocative): Two of the main conclusions drawn from this study and mentioned in the Abstract as point (1) and (2) (page 2, lines 22 to 28) can be easily derived without using any cluster analysis.

*The cluster analysis has been proven successful in many previous studies (cited in the manuscript, Dall´Osto et al. 2010-2018 and Beddows et al., 2009-2016) where the advantages and benefits were shown. In a nutshell the clustering analysis can greatly simplify the interpretation of aerosol size distributions. Indeed we show in page 6 line 8-20 two examples. Nevertheless, we briefly expanded the section and references.*

(ii) Moreover, from my point of view, it would be beneficial or even necessary to focus from case to case more on the original SMPS data, primarily when assigning air mass origins to NPF events. Here a more detailed discussion of air mass histories along with the original, individual PSD-spectra could be much more meaningful (the sketchily approach presented on page 14, lines 12 to 32 is hardly adequate).

*We expanded this section and added a NPF formation events and described it (see Figure SI9) and text in the section.*

In case of "Nucleation" cluster: Do the individual PSD-spectra show particle growth in contrast to the spectra assigned to "Bursting"? Especially here, you may present some examples from the original data set to demonstrate the unique characteristic.

*This paper has already 8 main Figures and 9 Supporting information Figures. We added into words two typical PSD-spectra (already described in many previous papers - Dall´Osto et al., 2017, 2018, 2019 and we feel we do not need to repeat them again). We added an example of a nucleation cluster episode (Figure SI9) and described in words that "bursting" are not defined events where grown is not seen (often also called "apple events").*

(iii) Air mass back trajectory analysis is a fundamental scaffolding of this study. The trajectory cluster analysis is interesting on its own but, however, somewhat detached from the PSD cluster analysis. I recommend presenting a figure analogous to Fig. SI 7 in the main text, but showing here trajectory ensembles sorted according to the PSD clusters as described on page 14, lines 12 to 32. Just another (minor) point concerning Fig. SI 7: The plot for cluster 1 (sea ice) shows terrain heights typically around 200 m or so, though the air masses travelled across the Weddell Sea (terrain height should be around zero!) – please check and clarify!

The calculation leading to the plot of C1 in Fig SI7 is correct.  The aggregated back trajectory starts at a height of 692m above the ground (terrain) and arrives at 10m above the receptor site.  We can't lower this without the trajectories grounding. But the average trajectory does not reflect the wide spread of height values.  The median trajectory height is around 300m with a lower 25th quartile has heights between 5 and 10.  This is further exemplified by a comparison between the histogram of step heights of the trajectories for cluster 1 and nominally cluster 6.  It is clear  from these that histogram of heights is heavily skewed towards low –level trajectories.

(iv) I recommend moving Figures SI 3 and SI 4 presented in the Supplementary Information (SI) to the main text, because they contain crucial information.

*We considered it, however Figure SI4 is basically a repetition of Figure SI2, and given we already have 8 main Figures in the manuscript, and we added to additional SI Figures (now 9 in total) we decided to leave them in SI material.*

(v) Whenever possible, provide corresponding uncertainties or standard deviations of the results, especially for any values given in "%" (regarding text and figures).

*Given average errors in legends, about 25% average.*

(vi) The pivotal question you raise addresses the balance between secondary vs. primary aerosol in this region (see Abstract lines 8 to 11 and p. 16, lines 2 to 4). I suggest picking up this quest in your conclusions more explicitly.

*We edited it and expand briefly the conclusions.*

Finally: Do you have any suggestion for future research on this topic?

*We edited and explained that further studies will analyze the SMPS data from multiple stations for the year 2015.*

Some specific and minor points:

1. Abstract: Please concretely state here size range, temporal resolution and measuring period.

*Edit. 6-209 nm in size range, daily resolution*

2. Page 4, line 29: . . .higher NPF instead of higher N.

*Edited*

3. Chapter 3.2: The association of PSD with meteorology, physical and chemical parameters appears rather descriptive. Do you have any ideas regarding the physical background of your findings?

We expanded this section

4. Page 9, lines 17 to 19: Hijman (2019) and Becker (2018) are not listed in the references.

*Edited and modified*

5. Page 14, line 19: Why did you relate to a total travel time of just 60 h and not 120 h (5 days back trajectories)?

*Based on previous studies (Dall´Osto et al., 2017, 2018) we chose to focus on shorter air mass trajectories to study new particle formation and bursting aerosol categories.*

6. Page 16, line 12: Please state "low particle number concentrations" more precisely.

*Edited, 121-279 particles cm-3*

7. Page 17, lines 5 to 16: I guess, during winter nss-sulphate aerosol should be negligible compared to sea salt. Maybe an additional closer look into the material presented in Rankin and Wolff (2003) or previous results on the chemical composition of the bulk aerosol from that site could be revealing, especially assessing the role of primary aerosol acting as CCN.

*We edited the text. However, it should be kept in mind that aerosol mass and aerosol number concentrations can be misleading - comparing PM mass with aerosol number concentrations may lead to wrong conclusions.*

8. Page 18, line 16: Please state "baseline" more precisely.

*Edited, annual average*

9. Figure 4, caption, line 9: . . . during the year 2015 (not: during the year 365).

*Edited*
The authors analyze particle size distribution data from coastal Antarctica using statistical methods to draw conclusions about aerosol sources and atmospheric processes. The results presented are both valuable and novel and are definitely within the scope of ACP. The context of the analysis and some of the actual discussion, especially as it relates to the existing literature, needs to be expanded but the necessary additions are minor with regards to the overall manuscript. Therefore, I recommend final publication with (generally) minor revisions.

*Many thanks for the appreciation of the paper*

Major comments:

Clustering analysis (especially S.2.2.1 and S.3.1.2) – The discussion on the clustering analysis needs to be greatly expanded as this is a fairly novel technique in atmospheric science. Many people fall into the trap of thinking that this machine learning method is actually machine intelligence and simply gives a "correct" answer as opposed to a mathematically valid solution. First, the values given for the Dunn Index and Silhouette Width need to be given context. Primarily, plots of both versus cluster number should be offered as many readers have no experience using or analyzing cluster analysis results. Secondarily, the values themselves need to be discussed in much greater detail. The 4x increase of the Dunn Index is good but 10ˆ-3 is still an extremely small value and implies that the clusters are extremely sparse (not compact), are not very far apart, or both. A graph of the cluster points to visually inspect both compactness and distance between clusters may be useful but may also be misleading as ambient data sets are often quite messy. Second, some additional validation of the cluster choice must also be presented.

*We edited all new section 2.2.1 and added two additional SI Figures (SI 1 and SI2).*

*We agree with the Referee 2 in that there is a potential trap to fall into, thinking that this machine learning method is actually machine intelligence and simply gives a "correct" answer as opposed to a mathematically valid solution. This is why we only use cluster metrics as a rough guide to inform our decisions. More often than not, our approach is to use a higher number of clusters and then manually aggregate them according to their shape and temporal trends. This in itself ensure that we do not miss any details and helps us select the optimum setting for the analysis to produce a result which*

*best describes the environmental conditions. Consequently, we do not give so much value to the cluster statistics that they drive the analysis towards a mathematically valid solution over a valid environmental solution. Hence, below we have given a description of the metric we have used and agree that we can put these into context.*

*With this in mind, the validation Indices used in this study were the Dunn Index (DI) and the Silhouette Width (SW) and the reader is referred to two very useful papers in order to understand these metrics fully (Halkidi et al 2001 and Rousseeuw 1987). The DI value at its most simplest can be thought of as the ratio of the minimum Euclidian distance between any two elements in two different clusters dmin to the largest separation of two elements within any of the clusters dmax. A DI value of 1 is obtained for a dataset with dmax = dmin, i.e. where the largest diameter of a cluster is equal to the minimum separation of between the circumference of any two of the clusters. Similarly, a DI value of 0 is obtained when dmin << dmax, i.e. if the clusters are touching each other. On this scale, our values of the order of 10-3 are indicative of clusters which are close to touching each other. This is not helped by the normalisation of the NSD which removes the total magnitude of the particles count from the data but it is important that we NSD shape without any bias due to the number concentration in the NSD. We also need to remember that this DI reflects the proximity of the two most similar clusters and does not reflect the separation of the other clusters which will be larger. In fact, this minimum separation is susceptible to outliers in the clusters.*

*The Silhouette Width is a measure of how well the elements i of a cluster matches the cluster (i.e. how well it has been classified) and a good description of this metric can be found in Halkidi et al 2001. A value close to 1 indicates that the elements are classified correctly and a value of 0 indicates that the elements ought to classified in a different cluster arrangement. Our value of around 0.4 (comparable to those discussed in Halkidi et al 2001) is the average value of the Silhouette Widths for all 8 clusters which ranges from 0.3 to 0.55. But to appreciate the quality of the clustering associated with the DI and SW values, Figure A presents the plots of the NSD for each cluster. Rather than a plot of points in an arbitrary space, we have use this to demonstrate the compactness and distance of the clusters. Each graph plots all of the daily average NSD for that cluster as a black curve and are compared with the average NSD plot of that cluster. From this figure, it is clear that we have good separation of the data for all of the clusters with the odd spurious NSD in clusters 1, 3, 4 and 7 which are not sufficient in number to form their own cluster but are allocated to their nearest cluster. It is these few sporadic NSD which lower the SW and DI values. From this optimum situation, it can envisioned that as we reduce the number of clusters we will loose the integrity of the separation and we might well expect the cluster elements to aggregate into larger clusters according to their modal diameter, eg Clusters 1, 3, 4 and 7; clusters 2, 6 & 8; and cluster 6. In fact, when we calculate the minimum standard deviation of the points about the mean NSD for each cluster setting, this is a minimum for 8 clusters.*

*We can illustrate this further when we plot the cluster number (as a colour) against the first 2 principle components (PC1 and PC2) in a bivariate plot. From this, we can see the grouping of the clusters (Figure B). This plot shows that there is no clear separation between neighbouring clusters and hence it*

*follows that the DI is more sensitive to the compactness of the cluster and when used with the SW find the optimum grouping of the NSD. The Nascent and Aiken (1 and 4) clusters group together on the bottom of the plot. Clusters 3 and 7 are on the top right hand of the plot and describe the Nucleation and Bursting and define to separate areas of the plot. The remaining 3 clusters, named Pristine (2, 5 and 6) fill in the gap between these two groups of cluster in the PC1-PC2 space. The remaining cluster called Bimodal forms the boundary between the two Pristine clusters 2 and 6. How this PC1-PC2 space is divided up as we increase the number of clusters from 2 to 8 can also be visualised in Figure C. At 2 clusters, the space is divided about a vertical line at just less than PC1-0.0 and as we increase this to 4, the data can convincingly be seen to divide into 4 equal spaces. As we increase from 4 clusters, we start to separate out the more interesting details in the NSD data at 8 clusters where we start to separate out types of nucleation and bimodal distributions.*

*We think this is sufficient to give confidence that our method has produce convincing results and we refrain from extending the study as recommended in the second half of the 'Major Comments:' to work on heavily curated data, leaving this to the possibility of a follow on methodology paper which is beyond the scope of this study. In fact, we are less inclined to include this under the bonnet/hood material discussed here in regards to figures A, B and C and use this in such a publication.*

[Figure]

[Figure]

Figure A. Daily normalised number size distributions plotted for each cluster (black lines) and overlaid by the mean number size distribution for the cluster.

[Figure]

Figure B. Bivariate plot showing the 8 clusters plotted as a colour agains the first 2 principle components PC1 and PC2.

[Figure]

Figure C. Bivariate plots showing the clusters plotted as a colour agains the

first 2 principle components PC1 and PC2.  This figure shows how the NSD are clustered as the number of clusters is increased from 2 to 8 clusters.

One way to do this may be to perform the analysis on heavily curated data to see if the results broadly match the overall analysis. I would highly (and very strongly) recommend that the authors run the analysis on a time frame where there are a minimum number of clusters expected (e.g. June). If there is broad agreement between the results there and the overall results, this would lend a great amount of strength to the overall conclusions. This could, and possibly should, be done in the context of air mass back trajectories as well where air masses could be broadly classified relative to their time spent over the continent, sea ice, or open ocean (this also comes with major caveats though, also see minor comment about back trajectory analysis).

*We addressed this above*

S.2.3 – is missing?

*Added*

S.4.1 – Much of the length of this section could be moved into the introduction and the remaining text expanded to give a more complete view of how these results fit into the existing literature. Overall, the authors do a fine job of finding relevant papers but do not necessarily discuss the conclusions presented completely. In particular, more discussion regarding measured composition and size distributions and the results presented here may be useful.

The results of Rankin and Wolff (2003), Preunkert at al. (2007, 2008), Saiz-Lopez et al. (2007), Schmale et al. (2013), Giordano et al. (2017), and many others should likely be discussed in greater detail. Additionally, the presence and lack of photochemistry should be given some context as this is a fairly dominating factor in the polar regions' winter vs. summer months.

*We tried to cite all the relevant papers, making sure the particulate matter mass and aerosol number concentrations could overlap. We edited all the papers and improved the text where possible. The majority of the studies report primary and secondary components in term of mass, which should not be confused with particle number concentration.*

Diurnal profiles (Fig. SI 3 especially) – The basis of this analysis, especially considering the weight the figure is given in the text itself, needs to be better justified. Diurnal profiles are generally helpful in visualizing the impacts of either photochemistry or timed anthropogenic activities (or both). Neither of these cases apply to the Antarctic continent. Either the analysis should be rerun in a more nuanced approach (e.g. diurnals for periods of 24-hours of sunlight and lack thereof, only run in the short timeframes of clearly demarcated sunrise/sunset) and discussed in that context or should be

removed completely. These results could be analyzed to give important insights into the potential role that the Polar sunrise/sunset plays in aerosol size distributions but this analysis may be beyond the scope of this manuscript.

*We do not have solar radiation data for this study. The impact of photochemistry may also be not straight forward as summer lacks of strong diurnal profiles. We agree this analysis is beyond the scope of this manuscript (already quite large given the 22 Figures presented).*

Minor comments:

P.7, S2.2 – A few sentences about the transmission efficiency of the aerosol stack for relevant sizes of aerosols should be added. The authors could consider applying a correction to the size distributions to account for inlet losses but I imagine they are fairly small for the relevant sizes.

*Indeed, as discussed in Jones et al., 2008 and in the text of this manuscript.*

S.2.4 and 3.3 – A few sentences regarding the accuracy of HYSPLIT being used in regions of sparse meteorological measurements should be added. A more detailed description of the initialization conditions for the model should also be added.

Back trajectories calculations are not renowned as being accurate, especially the further away in time and space from the receptor site. Fleming et al Atmospheric Research 104-105 (2012) 1–39 give a brief overview in their introduction. But we might well see a further reduction in predictive capability of the back trajectories if the metrological measurements are sparse.

Boccara et al 2008 consider this problem for lower stratosphere over Antarctica. https://doi.org/10.1029/2008JD010116. Simulated trajectories were computed starting from the positions of the balloon and advected using the ECMWF velocity fields. The simulated trajectories are compared to the real balloon trajectories. The spherical distance between the real and simulated positions exceeds 1000 ± 700 km on average after 10 days using ECMWF. We use 5 day trajectories but can use this to help us guage the accuracy at 5 days to be approximately 500 km which is just over the width of figure 4a. Clustered trajectories 1 is distinct from 2-6 due to its sole origin West of Halley. Whereas, the starting points of clustered trajectories 2-6 are all within this 500km uncertainty. Clearly, as the trajectory arrives at Halley, the uncertainty reduces to a minimum so we can have a higher confidence of the shape of the trajectories as they approach the receptor cite. With regards to the initialisation conditions: The meteorology files used are from the NCEP/NCAR Reanalysis Project is a joint project between the National Centers for Environmental Prediction (NCEP, formerly "NMC") and the National Center for Atmospheric Research (NCAR). The trajectories were collected with 10m, 30 and 60m arrival heights; we use 10m in the final analysis with 6 hourly steps. To compensate for the sparse meteorological

measurements, we calculate hourly trajectories and select those arriving at the hours: 00:00; 06:00; 12:00; and 18:00.

S.3.3 – The conclusions discussed in this section should be moved to S.4 and discussed in the context of the existing literature.

*We tried to edit and move the sections, but the results and the discussion sessions had to kept separate due the large amount of material presented.*

S.4.2 – The conclusions should be separated from the discussion. The work presented here is worthwhile and the main points should not be hidden.

*We expanded this section and the conclusion section explaining further studies will address this intercomparison study.*

Overall – consistency in figure references, especially for SI figures, should be double checked. E.g. Fig. SI 3 @ P.11 L11 vs Fig. SI3e @ P.11 L33).

*Edited*

Overall – references need to be double checked in both the main text and the references section.

*Edited*

References:

Rankin, A. M. and Wolff, E. W.: A year-long record of size-segregated aerosol composition at Halley, Antarctica, J. Geo-phys. Res., 108, D244775,doi:10.1029/2003JD003993,2003.

Preunkert, S., Legrand, M., Jourdain, B., Moulin, C., Belviso, S.,Kasamatsu, N., Fukuchi, M., and Hirawake, T.: Interannual vari ability of dimethylsulfide in air and seawater and its atmosphericoxidation by-products (methanesulfonate and sulfate) at Dumontd'Urville, coastal Antarctica (1999–2003), J. Geophys. Res.,112, D06306, doi:10.1029/2006JD0075857, 2007.

Preunkert, S., Jourdain, B., Legrand, M., Udisti, R., Becagli,S., and Cerri, O.: Seasonality of sulfur species (dimethyl sul-fide, sulfate, and methanesulfonate) in Antarctica: Inland ver-sus coastal regions, J. Geophys. Res. Atmos., 113, D15302,doi:10.1029/2008JD009937, 2008.

Saiz-Lopez, A., Mahajan, A. S., Salmon, R. A., Bauguitte, S. J. B.,Jones, A. E., Roscoe,H. K., and Plane, J. M. C.: Boundary LayerHalogens in Coastal Antarctica, Science,317, 348–351, 2007.Schmale, J., Schneider, J., Nemitz, E., Tang, Y. S., Dragosits, U.,Blackall, T. D., Trathan,P. N., Phillips, G. J., Sutton, M., andBraban, C. F.: Sub-Antarctic marine aerosol: dominant contri-butions from biogenic sources, Atmos. Chem. Phys., 13, 8669–8694, doi:10.5194/acp-13-8669-2013, 2013.

Giordano, M. R., Kalnajs, L. E., Avery, A., Goetz, J. D., Davis, S. M., and DeCarlo, P. F.:A missing source of aerosols in Antarctica – beyond long-range transport, phytoplankton,and photochemistry, Atmos. Chem. Phys., 17, 1–20, https://doi.org/10.5194/acp-17-1-2017, 2017.
Comments: Lachlan-Cope et al. present a novel study of Antarctic aerosol size distribution collected over a whole year of measurements at Halley research station. By applying the K-means clustering data analysis, eight aerosol categories were characterized. Based on the air mass back trajectory analysis, major sources regions including sea ice, open ocean, snow, and etc. were elucidated. Then, this study concluded that NPF and growth events in the Antarctic atmosphere mainly originated from both the sea ice marginal zone and the Antarctic plateau.

*Many thanks for the appreciation of the paper.*

The implication for climate and conclusion section is well-written, and in particular, the brief comparison with two other Antarctic stations (Dom C Concordia and King Sejong Station) during the year 2015 is a very useful and insightful section. Overall, the manuscript is generally well-written and interesting to read, with clear structure and sufficient explanations. The manuscript may be suitable to be published in Atmospheric Chemistry and Physics.

*Many thanks for the appreciation of the paper.*

Major Comments:

Page 2 and Lines 30: In the present study, cluster "pristine_160" with a bimodal size distribution (75 nm and 160 nm, respectively) shows the highest WS, but there were no correlations between them. Please clarify the meaning of "strong correlation in the abstract".

*Edited, associated*
Page 11 and Lines 19: New particle formation and growth was observed for the nucleation mode PSC cluster. In addition, the authors mentioned that NPF and growth events originate from both sea ice marginal zone and the Antarctic plateau in the abstract. Please, calculate and suggest the growth rate, which is a critical factor that affects the CCN number concentration in Antarctic regions. Then, the value could be compared according to the air mass origins.

Growth rate were calculated for the NPF events detected

Page 15 and line 9. Almost 5 pages were partitioned to discussion section that was overlapped with introduction section. Most of the discussion section should be moved to introduction section and SI.

*We tried to discuss our results and compare them with existing literature. That is why most of the more specific studies - for example primary and secondary aerosol sources - are discussed in the discussion section. The main issue is that we present here novel results from an entire year and we discuss aerosol sources and processes. We decided to leave a brief introduction section, and to provide a discussion of our results relative to the existing literature.*

Page 20 and line 19. The authors compared data from Halley, Dome C, and King Sejong Station. Overall, much higher concentrations are seen at the coastal Antarctic sites relative to continental based Dome C station. The coastal Antarctic stations being a remote location might be not immune to man-made impacts and specific tracers (e.g., black carbon) are necessary to discern those influence. In particular, quite high BC concentration was detected in King Sejong Station, as presented by Kim et al. (2018). Here, the possible sources of NPF and growth due to human activity (anthropogenic influence) could be discussed.

*We have a paper in preparation where aerosol size distribution data from the entire 2015 are analyzed all together. We edited the manuscript.*

Minor Comments:

Pate 9 and Line 1: Section 2.3 is missing.

*Edited*

Page 10 and Line 1: As mentioned in the manuscript, the difference between spring and autumn at Dp > 60 nm is very interesting. Please, explain possible reasons.

*The difference between spring and autumn at D>60 nm is also interesting, showing much higher concentrations in autumn, and likely due to a number of additional unknown sources including primary (sea spray) and secondary (sulphate and other components).Indeed clustering results in Figure 2 shows higher amounts of bimodal cluster and in general larger Aitken modes. Results are discussed in the discussion section.*

Page 2 and Lines 30: Error should be displayed in Figure SI 5. Please, provide the relationships between total particle number concentration and each meteoroidal data (e.g., wind speeds, RH, T, and ozone) according to the different aerosol categories.

*Error bars are generally 25%, not shown to emphasise different amoung different clusters.*

Page 38 and line 6: What is meant by "Please note that the sea ice extent is the median September extent from 1981-2010" in Figure 7.

*It is the median of the month of September, taken as average from the period 1981-2010.*

---

## Author Response (AR2)

Suggestions for revision or reasons for rejection (will be published if the paper is accepted for final publication)

The revised manuscript is a greatly improved version of the original one. I have a few, mostly minor, comments to be considered before recommending acceptance for publications.

*Thanks for appreciation*

Section 3.1.1 is a single, long paragraph. The text would be better readable if spit into 2-3 individual paragraphs. It is stated that "Results are broadly in line…". One should define which results. The results of this study? Some specific results discussed earlier?

*Done*

Also section 3.3 starts with a very long paragraph. To make this text more readable, I would again recommend splitting it into at least 2 paragraphs.

*Done*

The authors talk about minimum of the Hoppel mode (Abstract and end of section 3.1.1). I do not think this is quite a correct way of expressing this issue. In the scientific literature, this is usually called as a Hoppel minimum, and this minimum is then located between two modes, typically between Aitken and accumulation mode.

*Edited, Hoppel minimum*

There are a couple of small things related to sea spray aerosol in section 4.1.1. First, concerning the formation of sea salt by blowing snow, there is a very recent study (Frey et al. 2020, Atmos. Chem. Phys. 20, p. 2549) in strong direct evidence on this mechanism was obtained. This study could added to the citation list in this section.

*Indeed already cited, was under review. Frey et al 2019, now 2020.*

*Frey, M. M., Norris, S. J., Brooks, I. M., Anderson, P. S., Nishimura, K., Yang, X., Jones, A. E., Nerentorp Mastromonaco, M. G., Jones, D. H., and Wolff, E. W.: First direct observation of sea salt aerosol production from blowing snow above sea ice, Atmos. Chem. Phys., 20, 2549–2578, https://doi.org/10.5194/acp-20-2549-2020, 2020.*

Second, I am a bit skeptical in stating that coarse sea salt give a big contribution to a CCN population. Coarse particles tend to dominate sea salt mass concentrations but, being particle larger than 2.5 um in aerosol dynamic diameter by definition, their number concentration is simply too low to give a notable contribution to CCN. I would think the sea salt/sea spray population giving the largest contribution to CNN is centered well below 1 um, and probably close to 100 nm, in diameter.

*This is indeed a novel finding, as we show that these particles, much smaller than 2.5 um, may be related to blowing snow. Further studies are needed to support our observations, and are in preparation for submission.*

The reference list needs to be revised carefully. First, at least 3 references mentioned in the text (Draxler and Hess 1988, James 1989, Kyro 2013) are missing altogether from reference list. They might be others as well. Second, several of the references are incorrectly formatted. Please check out.

*Edited*

Finally, there a quite a few types in the text, especially in sections 1 and 2. Please check out the language of the very last version of the paper.

*Edited*